# An Islamic Perspective on Infection Treatment and Wound Healing

Tajwar Ali * and Haseena Sultan *

School of History, Zhengzhou University, Zhengzhou 450001, China
* Correspondence: tajooformanite@yahoo.com (T.A.); hasina_sultan@yahoo.com (H.S.)

**Abstract:** Muslims regard Islam as a complete code of conduct because it provides guidance in all aspects of life. Islamic teachings cover nearly all areas of knowledge, including medical sciences. Islam offers a unique perspective on how to treat wounds and illnesses. Islamic wound treatment methods are distinct and recognized by modern science. For Muslims, the only true treatment for illnesses and injuries is that mentioned in the Holy Quran and practiced by the Holy Prophet himself throughout his lifetime. Islamic treatments for various internal and external wounds and illnesses, such as the use of honey, black cumin, Indian incense, cupping, and cauterization, are extremely beneficial in treating both internal and external wounds. Islamic diets are high in beneficial nutrients for the body, such as ginger, figs, dates, and olive oil, and Islamic rituals such as five daily prayers, ablution, and fasting are very effective at keeping the body wound resistant. A healthy body has a strong immune system that can fight off various illnesses and injuries. To reach a definitive conclusion, a thorough examination of Islam's original and fundamental sources, such as the Holy Quran and the sayings of the Holy Prophet, was carried out. Although modern science has validated the majority of the approaches emphasized by Islamic teachings, much more research is needed to validate Islamic sayings about medical sciences.

**Keywords:** Islam; healing; wounds; diseases

## 1. Introduction

All religions around the world provide a set of guidelines and moral principles to help people live their lives methodically. Every religion is built on universal truths (Naik 2014a). According to Islamic teachings, (Quran 21:107) Islam is a religion that touches on practically every element of human existence and has offers comprehensive guidance on its legal, political, social, and economic facets. Islam offers a distinctive way of living, a humane diet, and medical care for both physical and mental health. The biggest source of Knowledge and guidance in Islam is the Holy Quran and the Sunnah[1] (Hamid 2003). Islam spread across the Arabian Peninsula in the seventh century, providing Muslims with both a highly intellectual and practical code of conduct. Islam has its own medical systems, and traditional Islamic techniques are very effective in treating wounds, injuries, and illnesses. Many Islamic ways and methods of treating ailments have been approved by modern science (Walbridge 1998). Muslims firmly believe that God created all ailments and that he also created the treatments for each one. The Holy Prophet of Islam said "There is no disease that God Almighty has created, except that he also has created its treatment" (Majah Vol. 4, Book 31, Hadith 3436). Muslims used different methods for healing cuts and wounds of war, as Islamic history is full of wars, victories, and wounds (Hitti 1970).

Despite the fact that Islam and Muslims have recently come under fire for fundamentalism, extremism, obscurantism, and, most importantly, terrorism, many experts have chosen to ignore Islam's positive aspects. Islam is a religion that offers its adherents a comfortable way of life. It has always advised Muslims to lead a healthy lifestyle. Additionally, it instructed those who followed them to treat their wounds, injuries, and cuts

using Islamic practices (Farage 2008). In addition to apparent incisions on a person's body, wounds can also be internal, mental, or psychological. Feelings and inner serenity can also be wounded. According to Islam, a person is comparable to a computer that combines hardware and software. The human soul is the software, while the body is the hardware. The soul is not a tangible thing, yet it does reside in the body. Islam emphasizes that the soul is the most significant and ever-living thing, whereas the body is only ephemeral. As software is crucial for a computer's performance, so is the soul for an individual's existence. Only one aspect is recognized by modern science, and that is the human body. According to Islam, the soul of a body can also be hurt, and it has a different treatment than the body. Islam regards psychological characteristics, moods, and other such aspects as being a part of the soul. Ibn e Sina explained that the body and soul are linked, but modern science does not acknowledge this. According to Islamic teachings, the soul is extremely essential because it may exist without the body and is inextricably bound to the body (Farage 2008).

Various studies have revealed that an individual's belief plays an important role in the treatment of a disease (Baghcheghi and Koohestani 2021). In Islam, one's belief system is an intrinsic component of one's soul; if a person has strong faith in a medicine's function and consumes it, the medicine is more likely to treat that person successfully. Belief is a software, and it is a kind of feeling which cannot be touched or manipulated physically. The soul is the major center of Islamic beliefs, with the body coming in second. It is believed that if an individual's spirit is at peace, his or her body will definitely be at rest as well. Offering five times salat, remembering God, reciting the Holy Quran, and other such good habits are encouraged in Islam because Islam considers these actions to be extremely beneficial to the soul. All such practices are healing the soul (the software) of human beings. In Islam, the soul and body have a vital connection (Sina 1959). The Holy Quran did not explicitly define numerous wound-healing treatments, yet this is the best healer for the soul, according to Islam. The Holy Quran specifically mentions honey as an effective wound healer and the Holy Quran as a wonderful healer for humans (Quran Chapter 16:Verse 68).

A wound is described as any damage or injury to the body's tissues, most often the skin. Wounds vary in severity and may be caused by a variety of circumstances, including cuts, punctures, burns, abrasions, or surgical wounds, while the invasion and growth of microorganisms (such as bacteria, viruses, fungi, or parasites) inside the body or wound site is referred to as infection. Infections may cause inflammation, slowed healing, and the spread of potentially hazardous germs. Healing is a complicated biological process aimed at repairing and restoring damaged tissues or wounds. It is characterized by a sequence of physiological processes that begin soon after the injury and persist over time. Inflammation, tissue formation, and remodeling are all part of the healing process (Healy and Freedman 2006). To heal the wounds of Muslims, Islam has traditionally used a variety of methods. Several practices were common in the Arabian civilization prior to the origination of Islam, but Islam only endorsed a small number of practices, and the Holy Prophet of Islam focused on specific techniques for treating wounds (Arozullah et al. 2018). Muslims throughout history employed various ways for healing wounds that were not mentioned in the Holy Quran and Sunnah, yet the Quran and Sunnah had established the guidelines for halal and haram. One of the methods for healing wounds was putting a burned mat of palm leaves and inserting the ash in the wounds. Later on, it was proved scientifically that ash is effective in treating the oral fungal infection, hemoptysis. Other methods of healing were the use of Indian incense because it was considered a treatment for more than seven diseases (AlRawi et al. 2017). Cupping and cauterizing were two other famous methods utilized by the Holy Prophet of Islam Hazrat Muhammad to cure different inner and outer wounds of ailing patients during his life time. Honey was applied to wounds of battles during the time of Hazrat Muhammad (PBUH) but later on it was proved scientifically that honey can kill the germs from wounds, and it is an effective healer. These methods are still in use in most of the Islamic countries as approved methods for the healing of wounds in Islam (Ghazanfar 1995).

The diet prescribed by Islam is also regarded as a super food diet with the power to heal wounds quite well. Honey is a healer in Islam for almost all kinds of wounds in the body of an individual. It can be used as both an external and internal medicine. It can be used as a food and as a cream for external wounds. Honey is mentioned in the Holy Quran as a powerful healer as part of the diet. All the abdominal diseases in Islamic history were cured through honey. Fruits such as dates, pomegranate, fig, olive oil, vegetables, and other food items which have been recommended by Islam are not only fast healers, but they have the ability to fight some of the deadliest and fatal diseases such as diabetes, cancer, heart attack, uric acid, and other many diseases. It is also the belief of Muslims that all these fruits are the fruits of paradise. Vegetable consumption has been encouraged in Islam, while meat consumption has been prohibited most of the time (Mayton 2010). Modern science has proved that all these food items are beneficial for human beings and these fruits and vegetable have the ability to increase the immunity of individuals (Dolatkhah et al. 2020).

Islamic practices are very significant for the healing of wounds of the human body. Fasting is a religious practice in Islam that can be beneficial for many stomach injuries because the stomach is considered the center of all ailments. Five times prayer practice in a day keeps the body fit mentally, physically, and psychologically. Ablution or the cleaning of the hands, mouth, nose, face, arms, head, neck, and feet of human body are necessary before performing five-time prayers for Muslims. In this practice, Muslims clean the important parts of the body more than five times in a day. This can prevent bacterial and fungal assaults on one's private parts. This practice also stops further injuries and it speed up the healing process of wounds. Prayer five times a day is a good exercise, similar to yoga, which keeps the body fit. In Muslim prayers, numerous postures are used; a person stands up and sits down several times. Other Muslim customs, such as washing the buttocks after using the toilet, have been scientifically confirmed to be healthful. When water is scarce, they use dry soil to clean their private parts, a practice known as dry ablution (Tayamum) in Islam. Ablution and Tayamum[2] has been stressed in the Holy Quran (Quran, Surah An Nisa 4:43). Washing private parts with water and dry soil is more beneficial than cleaning them with tissue paper according to modern science (Kamran 2018).

Muslims' healthy practices, tactics, and nutrition are beneficial for the rapid healing of wounds, and these ways are considerably more effective in the prevention of fatal diseases. In this paper, major Islamic traditional methods, Islamic diets, and Islamic practices that are effective for the fast healing of wounds have been elaborated with special reference to the most authenticated sources of Islam, such as the Holy Quran and the traditions (Sunnah) of the Holy Prophet Hazrat Muhammad (PBUH), as well as the opinions of some major figures in Islamic history. This study is a contribution in the field of religion based healing because Islam is the second biggest religion on the earth, which covers many aspects of human life. This study also contributes to the field of medicine and medical sciences because it shares the history of medical sciences in Islam.

## 2. Materials and Methods

To carry out this investigation, a thorough literature review methodology was employed. To gain a thorough understanding of the main research subject, original authentic Islamic sacred literature was examined, extensive reading of the Quran and authentic books of Muhamad's (PBUH) traditions was ensured, and research articles in Arabic, Urdu, and English were used. Because the Holy Quran is the main source of knowledge for Islam and Muslims, the study of it was carried out using a content analysis technique. In order to research authentic Islamic traditions on the treatment of wounds, the six books of Hadith, including Sahih Bukhari, Sahih Muslim, Ibn Sunan Abu Dawood, Sunan al-Tirmidhi, Sunan al-Nasa'i, and Sunan ibn Majah, and other explanations of the Holy Quran such as Tafsir al-Bayan by at-Tabariyy (310H), Tafsir Al-Kashshaf by al-Zamakhshariyy (467-538H), Tafsir Al-Muharrar al-Wajiz by Ibn e Atiyyah Al-Andalusiyy (481-442H), and Tafsir Fakhr al-Raziyy were examined. Islamic medical books such as Bihar Ul Anwaar and other such books written by early Muslim physicians were consulted for content analysis. The analysis

of Islamic scholars' lectures on the treatment of illnesses and the healing of wounds was performed using electronic sources. To gather primary data for this study, the lectures given by contemporary Islamic scholars including Dr. Zakir Naik, Dr. Tahir Ul Qadri, and Tariq Jamil on Islamic ways to treat wounds were carefully examined.

## 3. Islamic Traditional Wound Healing Methods

Various wound-healing treatments have been used throughout Islamic history. The great majority of Muslims, regardless of culture or region, accept the essential Islamic methods provided in the Holy Quran and taught or authorized by the Holy Prophet of Islam Hazrat Muhammad and other precious Islamic books. These traditional approaches rely on the Quran and Sunnah for knowledge and wisdom. They stress the significance of hygiene, the use of natural substances such as as honey and black seed oil, and the recitation of specific scriptures or supplications for healing.

Throughout history, Muslims have used these traditional treatments to treat a variety of wounds, from cuts and burns to bruises and surgical wounds. Individuals seek balance and improve their bodily and spiritual well-being by embracing these ancient healing approaches. Finally, Islamic traditional wound healing treatments provide a distinct healing strategy that blends natural medicines, prayers, and the direction of the Quran and Sunnah. They are an essential element of Islamic culture and are used in conjunction with current medical methods to treat both the physical and spiritual aspects of recovery (AlRawi et al. 2017). The types of wounds in islam and their treatment has been explained in Table 1.

**Table 1.** Types of wounds and the proposed methods of healing from Quran and Sunnah.

| Types of Wounds | Proposed Healing Methods |
| --- | --- |
| Cuts and Grazes | Applying honey and using blackseed oil |
| Burn Injuries | Applying cold water, ash of palm leaves |
| Bite or Sting Wound | Rinse the wound with water |
| Bruises | Applying a cold compress |
| Internal Injuries | Seeking immediate medical attention |
| Gunshot or Stab Wounds | Applying ash of burned palm leaves |
| Surgical Wounds | Keep the wound clean and sterile |
| Illness | Recite specific dua'as for healing (Ashy 1999) |

### 3.1. The Super Method

In Islam, the best and most effective method for healing all types of wounds is the one emphasized in the Holy Quran. The Quran is the cure for all ailments.

Allah says in the Holy Quran that "O mankind! There has come to you a good advice from your Lord (i.e., the Quran), and a healing for that which is in your hearts." (Quran 10:57)

شَرابٌ بُطونِها مِن يَخرُجُ ۚ ذُلُلًا رَبِّكِ سُبُلَ فَاسلُكي الثَّمَراتِ كُلِّ مِن كُلي ثُمَّ

يَتَفَكَّرونَ لِقَومٍ لَآيَةً ذٰلِكَ في إِنَّ ۗ لِلنّاسِ شِفاءٌ فيهِ أَلوانُهُ مُختَلِفٌ

The Holy Quran is healing for all those who believe in it according to Islamic teachings. The Holy Quran devotes a great deal of attention to the spiritual side of human beings, and it is believed that reading the Holy Quran can help bring peace to an individual's soul. The Holy Quran was revealed to humanity in a poetic form, and the act of reading the verses of the Quran has the effect of making one's ears more pleasant. The concept of healing is rooted in the Holy Quran. The Quran has declared the entire Quran to be a healing for

humans. This verse also indicates that whatever the Holy Quran discusses is the cure for all diseases that modern medical researchers have yet to identify (Martin 1982). The Holy Quran is written in a highly symbolic language that is difficult for the general public to comprehend, but men of understanding can decipher its meanings by reading between the lines (Stowasser 1995). As for physical healing, methods in Islam are concerned the Holy Quran directly expressed the importance of honey. God has mentioned honey as a good healer for many diseases in the Holy Quran. In Verse 69 of An Nahl, Allah says: "From within their (i.e., the bees) bellies comes forth a fluid of many hues, that provides people with a cure (of illnesses)" (Quran, 16: 69) This verse of the Holy Quran certifies that honey is an active healer for almost all diseases because it is made by a fluid of many hues. Alvarez-Suarez and co-workers approved this in their report and stated that antimicrobial activity is present in all types of honey (Alvarez-Suarez et al. 2010). Modern science has found that honey is antibacterial and has an inhibitory effect on many organisms which are dangerous for human beings. Table 2 shows few of the bacteria and organisms sensitive to honey, but there are a myriad of bacteria that are sensitive to honey.

**Table 2.** List of organisms and bacteria sensitive to honey (Molan 1992).

| Actinomyces Pyogenes | Pseudomonas Aeruginosa |
| --- | --- |
| Bacillus anthracis | Salmonella cholerae-suis |
| Campylobacter coli | Streptococcus agalactiae |
| Campylobacter jejuni | Staphylococcus aureus |
| Candida albicans | Salmonella typhimurium |
| Corynebacterium diphtheria | Shigella species |
| Echinococcus parasite | Serrata marcescens |
| Enterococcus avium | Salmonella typhi |
| Enterococcus faecium | Rubella virus |

### 3.2. Method 2: Prophet Muhamad (PBUH)'s Methods

During his lifetime, the Holy Prophet of Islam used numerous methods to heal the wounds of Muslims, and his practices and traditions are followed by nearly 1.8 billion Muslims worldwide today. Before the advent of Islam in Arab society, a number of well-known healing practices were sanctioned by the Holy Prophet. He used his own intelligence and employed specific methods and medicinal foods for healing, which are still widely known as Hazrat Muhammad's methods (Tib e Nabvi). Although the Holy Prophet was not an expert in the medical field, he was illiterate (ummi), he had a divinely blessed wisdom according to Muslim beliefs (al-Quḍāʾī 2016).

### 3.2.1. Method I

Using honey to treat several diseases was another method favored in Islamic history. In the Holy Quran, the importance of honey is stressed and the Holy Prophet suggested honey for abdominal wounds and troubles. Use of honey is a method suggested by God, and the Holy Prophet further suggested to apply honey on wounds, according to an Islamic tradition.

"A man came to the Prophet and said, 'My brother has some Abdominal trouble'. The Prophet said to him 'Let him drink honey'. The man came for the second time and the Prophet said to him, 'Let him drink honey'. He came for the third time and the Prophet said, 'Let him drink honey'. He returned again and said, 'I have done that 'The Prophet then said, 'Allah has said the truth, but your brother's' Abdomen has told a lie. Let him drink honey.' So, he made him drink honey and he was cured" (Al-Bukhari 2020a).

Honey contains numerous minerals, amino acids, vitamins, and antioxidants. Honey contains the vitamins niacin, riboflavin, and pantothenic acid, as well as the minerals

calcium, copper, iron, magnesium, manganese, phosphorus, potassium, and zinc. Additionally, it contains a variety of flavonoid and phenolic acids that function as antioxidants, preventing aging and eliminating free radicals. Honey has a long history in human nutrition and is used as a sweetener and seasoning in a variety of foods and beverages. In the Quran, honey is described as the best source of healing, and it is also listed as one of the foods of paradise (Purbafrani et al. 2014).

The modern scientific community acknowledges honey's efficacy in treating wounds and diseases. Honey is an amalgamation of plant juices, and honey bees manipulate this liquid. This natural object is well-liked due to its high nutritional and preventative medicinal value. It possesses potent antibacterial properties and is effective at preventing and eliminating wound infections. It has been used as a wound care product, and its use as a wound healing agent has been reported for the treatment of venous leg ulcers, burns, chronic leg ulcers, pressure ulcers, and catheter exit sites (Abou El-Soud 2012).

Honey is a non-traditional medicine that has been demonstrated to have the same or better therapeutic effects as conventional treatments. It is a cost-effective natural agent that does not pose any health risks and has the capacity to heal diabetic wounds quickly. It is an alternative to sugar that has properties that are antibacterial and anti-fungal. It has the ability to kill unwanted fungus and bacteria and prevent further infection and promote healing. Honey has antibacterial properties and is high in antioxidants, both of which help reduce the risk of cancer and heart disease (Fahmida Alam et al. 2014).

### 3.2.2. Method II

Cupping treatment is an alternative medicine in which a therapist puts special cups on one's skin for a few minutes to generate a pull and suction. It is used for different purposes, including blood flow, relaxation, and well-being, to help with pain, inflammation, and as a type of deep-tissue massage. It was an ancient method in the Arab culture which was approved by the Holy Prophet, and it was used by the Muslims widely and still many Muslims use this method (Zeng and Wang 2016).

The Holy Prophet approved this method in the Hadith, which is narrated by Ibn 'Abbas:

The Prophet said, "Healing is in three things: A gulp of honey, cupping, and (Cauterizing) but not with fire." But I forbid my followers to use (cauterization) branding with fire (Muhammad b. Isma'il al-Bukhari, Volume 7, Book 71, Number 592 2020).

The Holy Prophet also approved cupping as a good method for treatment of many diseases. During the time of the Holy Prophet, cupping was used to treat several diseases. Cupping is a scientifically approved method which had been used by many societies in history and is still being used in many cultures (Mehta and Dhapte 2015).

### 3.2.3. Method III

Another method which was approved and used by the Holy Prophet of Islam to heal the inner wounds was cauterization. This method was approved but it was not a preferred method of the Holy Prophet. The Holy Prophet allowed this method, but did not encourage it to be used frequently, or it was not preferred by him. In the above hadith, cauterization is also approved but not with fire. It is used to cure a wide range of ailments, such as toothache, stomachache, jaundice, eyesight issues, and muscle pain. The internal infections in the human body create all these health issues and these internal infections are also called the inner wounds which can be cured by cauterization as a Muslim traditional method. In this method, parts of the body are cauterized with a red-hot metal. In many Muslim countries of the world, such as in Oman, this method is still in use and popular. In Oman, it is still believed that this method works where modern science fails (Ghazanfar 1995).

### 3.2.4. Method IV

In the books of Imams Al-Bukhari and Muslim, the Holy Prophet teaches about how to stop bleeding with some traditional remedies. In the battle of Uhud, the Holy Prophet sustained facial wounds and a broken tooth. It is said that ash from a mat of palm leaves

that had been burned and inserted into his wounds stopped his bleeding. In this manner, wound bleeding was stopped. Inserting the ash of a burned mat of palm leaves is a method to stop bleeding in Islam. The ashes of burnt palm leaves are very effective in stopping bleeding, because it is a durable drying agent and because it has the slightest burning effect (on the exposed skin). Other robust drying medications have a burning effect on the skin and cause the blood to be disturbed and the bleeding to increase (Al-Jauziyah 2003). According to research in modern science, the healing of wounds in the skin relies on the availability of trace elements or enzymes which increase the structural components in the repair of tissues. Ashes contain all the required elements for the healing of wounds. Ashes also have the combination of metals which are needed to heal wounds quickly (Shaikh and Shaikh 2008).

### 3.2.5. Method V

Applying black cumin to wounds or utilizing it in other ways is the most effective method for treating wounds. In various hadith, the Holy Prophet emphasized the significance of black cumin and regarded it as the best remedy for almost all diseases, excluding death. Black cumin is the most effective healer, recuperative agent, and immune booster. According to The Holy Prophet of Islam, five or seven black cumin seeds should be crushed and combined with oil. The mixture should then be placed in each nostril (Muhammad b. Isma'il al-Bukhari 2020). Table 3 shows the importance of black cumin according to modern science.

**Table 3.** Scientific findings about black cumin.

| Major Nutrients | Health Benefits |
|---|---|
| Manganese, Mn 8.53 mg (370.87%) Copper, Cu 2.6 mg (288.89%) Iron, Fe 9.7 mg (121.25%) Total Fat (lipid) 31.16 g (89.03%) Phosphorus, P 543 mg (77.57%) Magnesium, Mg 265 mg (63.10%) Calcium, Ca 570 mg (57.00%) Zinc, Zn 6.23 mg (56.64%) Protein 22.8 g (45.60%) Potassium, K 808 mg (17.19%) Total dietary Fiber 6.03 g | • Weight loss, Protects the Gut • Suitable for Women • Anti-fungal Activity, Antiviral • Reduces Seizures, Boosts the Immune System, Treatment for MRSA • Protect against Heart Disease • Anti-Diabetic, Digestion • Protects the Kidneys and Prevents Kidney Stones (Yimer et al. 2019) |

The importance of black cumin was studied by many scientists and it has been proved that it is an excellent healer for different wounds and illnesses. Black cumin, a decidedly valued nutraceutical herb with a varied collection of health benefits, has attracted rising interest from health-conscious folks to scientific and pharmaceutical forums. The pleiotropic pharmacological properties of black cumin, and its chief bio-active constituent thymoquinone (TQ), have been demonstrated by their capability to weaken oxidative stress and inflammation, and to promote immunity, cell endurance, and energy metabolism, which trigger diverse health benefits, including defense against metabolic, cardiovascular, digestive, hepatic, renal, respiratory, reproductive, and neurological disorders, cancer, and so on. Furthermore, black cumin works as an antidote, extenuating various toxicities and drug-induced side effects (Hannan et al. 2021).

### 3.2.6. Method VI

The Holy Prophet of Islam Hazrat Muhammad also suggested to inhale Indian incense for those who have throat troubles or infections in the throat. This is also recommended to be put into one side of the mouth of one who is suffering from pleurisy. The most common symptom of pleurisy is a shrill chest pain when inhaling deeply and the pain sometimes is also felt in the shoulder. The Holy Prophet further stressed the importance of Indian

incense is that it can cure seven diseases (Al-Bukhari 2020c, Volume 7, Book 71, Number 600 2020).

3.2.7. Method VII: About Pandemics

Prior to 1400 years ago, the Holy Prophet of Islam provided comprehensive guidance regarding contagious diseases, which are regarded as the most effective methods for containing the domino effect of plagues, leprosy, epidemics, endemics, and pandemics. The Holy Prophet said, "If you hear of an outbreak of plague in a land, do not enter it; but if the plague breaks out in a place while you are in it, do not leave that place (Al-Bukhari 2020b)".

The best way to counter a plague is to limit the mobility of people from the affected areas. This method was widely used by many countries of the present world during the initial days of COVID-19 in early 2020. The Holy Prophet of Islam had also mentioned leprosy as one of the biggest contagious diseases, so that its prevention is the best healing of it. The Holy Prophet said that,

"one should run away from the leper as one runs away from a lion (Muhammad b. Isma'il al-Bukhari, Volume 7, Book 71, Number 609 2020)".

According to the Holy Prophet, the most effective treatment for contagious diseases is to avoid infected individuals. Plagues cause infections in the human body in the form of viral attacks on specific parts of the human body; this virus is highly contagious. The greatest difficulty in plagues is preventing virus transmission. The modern world was unable to invent a cure for the Corona virus, but in 2020, the Holy Prophet's method for preventing the virus's transmission was used once again.

## 4. The Impact of Islamic Diet on Wound Healing

As the saying goes, "You are what you eat", and similarly, food shapes the human body, immunity, and vitality. Islam provided comprehensive diet and nutrition advice. Diet as practiced by the Holy Prophet of Islam and the foods he ate throughout his life constitute an Islamic diet and the ideal food for Muslims. Healthy eating means having a healthy and ample diet that is rich in vitamins, vegetables, and fruits, as well as receiving enough nutrients from foods such as milk, cereals, meat, and beans. It also brings vitality and radiance to the skin and increases the human immune system's strength (Tsugane 2020). Food and diet are the major components that give immunity and support healing of wounds. For the healing of wounds, diet plays a vital role. External medication only stops the intervention of viruses and bacteria in the wound, but diet heals the body from the inner side of body.

Muslims categorize food into two groups: one is allowed food (Halal)[3] and the other is Haram[4] or forbidden. Halal food has been identified by the Holy Quran and The Holy Prophet of Islam. God says in the Holy Quran that,

"Forbidden to you (for food) are: dead animals-cattle-beast not slaughtered, blood, the flesh of swine, and the meat of that which has been slaughtered as a sacrifice for other than God..." (Quran 5:3) ...and intoxicants (Quran 5:91–92)."

Due to certain logical considerations, Islam prohibits the consumption of the flesh of the majority of animals. The modern scientific community concurs with a number of Islamic tenets regarding the effects of Haram foods and the advantages of Halal foods. Modern science is still investigating a number of reasons for the prohibition of specific foods. All permissible foods and types of meat, such as lamb, chicken, and certain fishes, are nutrient-dense and considered clean. All foods permitted by Islam are renowned for their medicinal value and rapid healing properties. Table 4 lists animals whose flesh is permissible to consume and those whose flesh is not. This table also includes the scientific rationale for permitting or prohibiting a practice.

**Table 4.** Halal and Haram Meat in Islam.

| Food | Allowed | Not Allowed | Scientific Reason |
|---|---|---|---|
| Pork | No | Yes | Poisonous and filthy flesh |
| Mutton | Yes | No | High proteins and iron |
| Beef | Yes | No | Proteins, vitamins, minerals |
| Donkey | No | Yes | Islam considers it unclean[5] |
| Pets | No | Yes | Rabies and bacterial infection |
| Some Fishes | Yes | No | Pathogenic bacterial hazards[6] |

Eating dead animals which are not slaughtered is strictly prohibited in Islam because they can be harmful for human health, which was also proved by modern science later on. When the blood of animals does not leave the body as it should and instead becomes adjusted in the meat, this can be the cause of many diseases in human beings. According to research, blood is a good medium for bacteria, germs, and toxins. The Muslim way of slaughtering is more hygienic because it eliminates most of the blood containing toxins, germs, and bacteria that can cause various diseases (Haque et al. 2018). The meat of slaughtered animals is free from toxins and bacteria, and it cannot be an obstacle in the healing of wounds; rather, it precipitates the process of the healing of wounds. The meat of unslaughtered animals can further worsen the situation of wounds.

It is also forbidden in Islam to consume the flesh of pigs because it is believed that eating pork can lead to a variety of health issues in humans. Because it is so high in fat and causes the body of the eater to produce more fat, swine flesh is implicated in more than seventy different diseases, according to the most recent scientific research. According to one theory, eating pork raises the level of erotic behavior in humans to such a point that they lose morality (Naik 2014b).

In the Holy Quran, Allah has recommended that "*eat what is lawful and good in the Earth* (Quran 2:168)". Searching through the Quran further, we can identify what foods are beneficial; these include honey (Quran 16: 68–69) vegetables such as corn and herbs (Quran 55: 12), and fruits such as olives, dates, grapes, pomegranates (Quran 6: 99–141), and bananas (Quran n.d.). All of the foods recommended by Islam are scientifically proven to boost the immune system. Increasing immunity increases the rate at which wounds heal. All the mentioned foods and fruits are full of energy and they have a high medicinal value. Fruits mentioned in Figure 1 are also considered as the fruits of heaven by Muslims. Islam gives a special importance to these fruits.

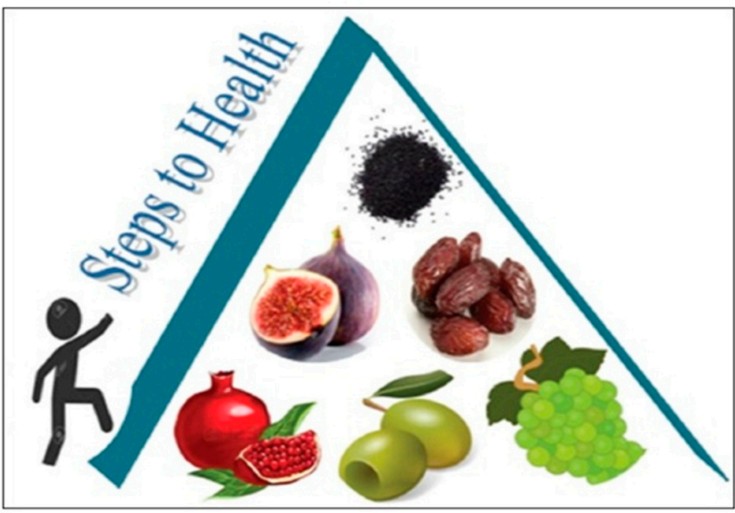

**Figure 1.** Islamic recommended fruits for many diseases.

He has also recommended us to eat the meat of certain animals and their milk, as well as fresh fish and birds. The Holy Quran discourages eating more than necessary, which creates huge problems in the human body: "*eat and drink and do not commit excesses; indeed, He does not love those who are excessive*" (Quran 7:31). In Islam, overeating is strictly prohibited because it can cause a multitude of problems for humans. Overeating is detrimental to one's health, as it can invite more diseases into the body. In Islam, abstinence and the avoidance of toxic foods have been heavily emphasized. In Islam, a man who obtains and avoids all toxic foods is considered pious. As the adage goes, prevention is preferable to treatment, and the same is emphasized in Islamic teachings. The Holy Prophet said that "Whoever eats little, will have a sound body and a pure heart, and whoever eats too much will get sick and will become hardhearted (Majlisi n.d.)".

The sayings of the Holy Quran and the Holy Prophet regarding food and diet are the most effective methods for wound healing. Healing is the process of recovering from illness, and this process requires humans to consume a healthy diet. A healthy diet fosters an immune system that promotes healing. The use of the Islamic diet and Islamic methods of using diet can aid in the recovery from illness and the healing of wounds. Different foods were consumed by the Holy Prophet and his companions during his lifetime, and these foods are currently regarded as the best in the Muslim world. The modern scientific community also recognizes these foods as rapid healing agents. The Holy Prophet explained the significance of numerous foods and their medicinal properties. The Holy Prophet, while explaining the importance of Ajwa dates, said that eating seven dates in the morning can save one person from the attack of poison and magic in that day when he has eaten dates (Muhammad b. Isma'il al-Bukhari 2020).

Dates are a staple food item in Middle Eastern culture and are extremely popular in almost all Islamic nations. The Holy Prophet recommended eating dates while breaking a 12 h fast. In this manner, dates can be considered an instant source of energy, as well as an instant healer. Dates were the Prophet Muhammad's (PBUH) favorite food, as he stated, "If anyone is fasting, he should break his fast with dates. If he does not have them, he should use water (Al-Farsi and Lee 2008)".

Scientific studies also reveal that the date palm (P. dactyliferous L.) is a multi use tree, providing fiber, carbohydrates, minerals, and vitamins, besides having extreme medicinal properties (Hanano et al. 2017). Polyphenol-containing dates were found to have anticancer effects, as reported by the study of Eid et al. (2014).

Ginger is another food item which is used by the Muslims as an excellent healer of wounds. It can be applied to the wounds, and it is also used as a food to heal wounds. In Islam, ginger has been considered as an important food item. Its importance has been mentioned in the Holy Quran, and it has been explained by the Holy Prophet of Islam as an excellent food item which can heal wounds. In the Holy Quran, ginger is mentioned in this way: "And they will be given to drink a cup of (wine) mixed with Zanjabil (ginger) (Quran, Surah Insan 17 n.d.)". The dried form of ginger is applied to the outer wounds and fresh ginger is used in cooked food by many Muslims, considering it as the best medicine for many diseases. Modern science also discovered the importance of ginger and considers it to have healing properties against cancer. The anticancer potential of ginger is well documented and its practical constituents such as gingerols, shogaol, and paradols are the valued elements which can halt numerous kinds of cancers (Mashhadi et al. 2013).

Allah Almighty has mentioned the importance of fruits such as grapes and dates and called them the most important diets. God says, "And from the fruits of date palms and grapes, you obtain (date and grape juice) and a goodly provision. (Quran Surah Nahl:67)." Grapes and dates are the best fruits and they are very high in nutritional value.

God emphasized the significance of figs and olives in the Holy Quran by swearing on them in Surah At Tin. It is clear that Allah Almighty values his own creativity in this surah. Since adherents of the Abrahamic faiths hold that God created everything in the universe, including humans. Allah says "By the Fig and the olive" (Quran Surah At Tin:1) Surely a fruit that Allah swears by, is tremendously advantageous for human health. The

Holy Prophet also considered Olive oil as the treatment of many human diseases. The Holy Prophet said that "Eat the olive oil and apply it (locally), since there is a cure for seventy diseases in it, one of them is Leprosy" (Muhammad b. 'Isa al-Tirmidhi Tirmizi:1851; book 25).

Eating and applying of olive oil has become a Muslim tradition and Muslims consider this treatment as a most reliable one. Leprosy is a contagious infectious disease which can be healed by the use of olive oil according to Islamic teachings. Olive oil is also applied to the wounds of leprosy, and it is also an excellent food which can heal many inner wounds.

Figs and olives are revered by Muslims, not only for their heavenly flavor, but also for their ability to boost the body's immune system and hasten the healing of wounds. The Holy Prophet said, "If I could say that a fruit was sent down from heaven to earth "I would say it is fig because the heaven's fruit has no stone, eat it as it cures hemorrhoids and it is useful for treating gout" (Narrated by Abu Darda). Figs are considered good to reduce blood pressure, which in turn decreases the risk of heart disease and stroke. Figs are also very high in fiber. Figs help to get rid of sleep disorders such as insomnia and also treat chronic constipation and several other diseases according to modern science (Arvaniti et al. 2019).

The prescription of a huge number of herbal foods such as pomegranate, olives, figs, dates, grapes, and black seeds was considered medicine by the Holy Prophet of Islam. Recently, these have become super foods with their prevailing healing properties and act as advantageous dietary interventions for disease treatment, healing, and maintenance of health. The use of these foods as constituents of natural sources with fewer side effects seem to be more promising than chemical treatments. Prophetic commendations of food are outstanding for their clairvoyance, as they came centuries before research was led on healthy diet and their benefits to the body (Deuraseh 2006).

## 5. Healing of Wounds through Daily Practices of Islam

A highly practicing Muslim enjoys a busy daily routine. Modern science considers all of his actions in one day to be healthy habits. For example, a practicing Muslim will go to bed much earlier than usual because he must rise for morning prayers. He is an early riser, so he sleeps earlier at night. Because he must perform prayers five times per day, he performs ablution five times per day. Muslims are prohibited from praying prior to ablution. Daily, he cleans his teeth with Miswak, a stick of a specific tree that is beneficial for teeth. A practicing Muslim observes a period of fasting once every 12 months. He always remains mindful of God in every circumstance. Then, he eats, drinks, performs his duties, and attends to other matters in accordance with Islamic law. To perform Muslim prayers, one needs to move through numerous distinct corporal postures while narrating a specific supplication. It includes a certain level of physical activity which comprises standing, bowing for prostration and sitting sequentially. Each position involves the movement of diverse parts of the human body in ways that some muscles contract isometrically and some contract in approximation. The actions of prayer have the capacity to improve the flexibility and muscular fitness of the human body. This is the most effective form of physical exercise when performed five times per day. These movements in Muslim prayer can serve as a substitute for yoga and Pilates, as yoga is beneficial for the human body and rehabilitates numerous illnesses (Kumari and Pathak 2021). Figure 2 shows the similarities of the postures of Muslim prayers with yoga.

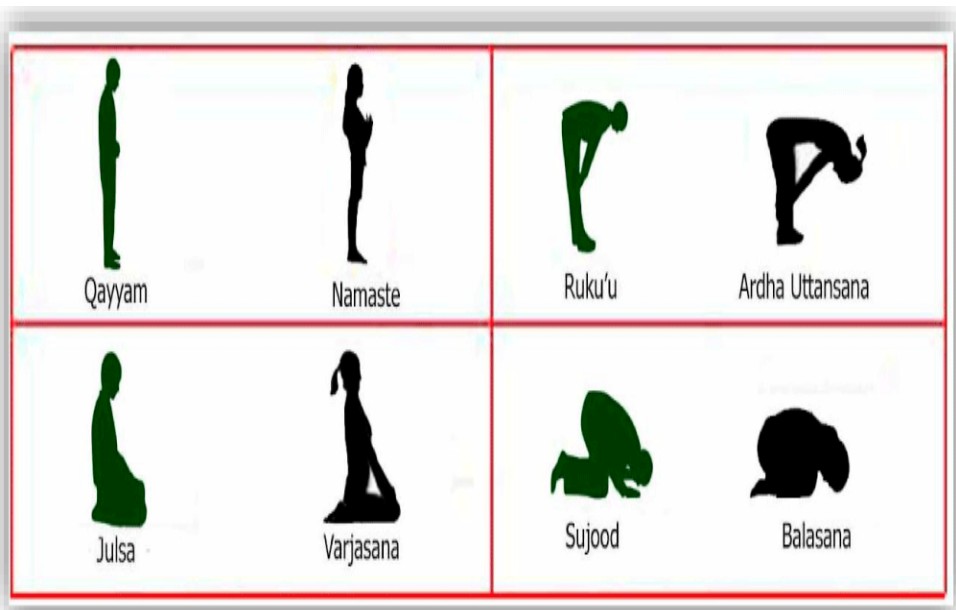

**Figure 2.** Similarities between Muslim Prayers and Yoga.

A highly Practicing Muslim repeats these Rak'ah[7] (movements) a minimum of 17 times every day. The number of total Rak'ah is approximately 48 Raka'ah per day which includes all mandatory and optional Rak'ah. Each Rak'ah consists of a series of 7–9 postures. Each posture moves muscles in a certain direction. Table 5 shows the number of postures and Rak'ah of a practicing Muslim in one day and in a week.

**Table 5.** The Number of Postures and Rakkat of a Practicing Muslim.

| Prayers | Number of Rakah | Number of Postures | Total Posture in 1 Month |
|---|---|---|---|
| Dawn prayer | 4 | 36 | 1080 |
| After Noon prayer | 12 | 108 | 3240 |
| The Late After Noon Prayer | 8 | 72 | 2160 |
| The Evening Prayer | 7 | 63 | 1890 |
| The Night Prayers | 7 | 153 | 4590 |
| **Total** | **48** | **432** | **12,960** |

In the same manner, Muslim prayers are better for the human body's fitness and it aids in the recovery from many psychological illnesses (Bradshaw et al. 2008). Fitness of the body increases the immune system's power, and an enhanced immune system can best heal the wounds (Kamran 2018). Prayers strengthen both the mind and the body. Many problems related to human psychology can also be resolved through the practice of prayers because it strengthens an individual's belief system.

Islam has always placed a premium on cleanliness, purity, and hygiene. In Islam, cleanliness is regarded as half faith. In this regard, Islam is the only religion on the planet that emphasizes cleanliness even at the most basic levels, which are often overlooked by followers of other religions. Islam has taught a clear method of cleanliness following waste secretion from the body, as well as the methods of bathing (Gusal)[8] and ablution for those who must offer prayers. It is obvious that physical cleanliness is the best method for fast wound healing, and Muslim cleanliness practices aid in wound healing. While a clean body protects wounds from bacteria, a filthy body welcomes bacteria and diseases and can make the existing wounds even worse. Islamic practices help the wounds to be healed in a fast manner. Allah has stressed in the Holy Quran about cleanliness that, "Truly, God loves

those who turn unto Him in repentance and loves those who purify themselves (Quran 2:222)". Faith in Islam is an unshaking power which has the ability to command the wishes of the individual. Considering cleanliness as half faith simply means it has a special status in Islam which has not been given to cleanliness in any other religion. The Holy Prophet Muhammad, (PBUH) declared limpidness to be half of faith. "Purity is half of faith, and the praise of Allah fills the scale (al-Naishapuri Sahih Muslim Book 2, Number 0432)".

Islam has guided the methods of ablution and other cleanliness methods that are hygienically good for human bodies and the prevention and treatment of diseases. In the Holy Quran, God teaches the method of ablution before prayers.

"O you who believe, when you get up to observe the Salat, you shall wash your faces and your arms to the elbows, and wipe your heads and your feet to the ankles (Quran 5:6)". Another alternative of ablution is also shown in Islam, which is considered as dry ablution or "Tayamum". In this method, Muslims are called to perform ablution with dry soil. Dry soil can also create purity, and this was told to the Muslims even 1400 years ago in the Holy Quran (John 1988)."And if you are ill, or on a journey, or one of you comes after answering the call of nature, or you have been in contact with women (by sexual relations) and you find no water, perform Tayammum with clean earth and rub therewith your faces and hands (Tayammum)" (Quran al-Nisa' 4:43).

Another healthy practice in Islam is fasting, known as soum or roza[9]. Muslims observe a period of fasting in one month in a year. Fasting has several physical benefits and it can also provide a period of rest for the digestive system of human beings.

Marriage is a highly encouraged institution[10] in Islam, which can help men avoid different dangerous diseases. When this practice is not encouraged, or it is made complicated, then youngsters will be attracted towards illegal relations with others. HIV Aids is typically transmitted from one person to another through sexual contact with different people. Marriage also reduces insecure anal sex behavior among heterosexual intravenous drug users, which becomes a complicated issue for human health (Mirabi et al. 2013).

Another study found that circumcision, "Khatna"[11], a compulsory act practiced by all Muslim males, has the potential to reduce HIV transmission from an infected female to uninfected male partner (Weiss et al. 2009).

It is an admitted fact that the lifestyle of an individual is responsible for myriad health complexities, and the same is responsible for a healthy life. It has been recognized that more than 5% of the causes of death are relevant to lifestyle. Health problems such as cardiovascular diseases, obesity, cancer, and diabetes type II in developing countries are deeply associated with lifestyle changes (Baguley et al. 2020).

The World Health Organization (WHO) has raised the idea of using religious principles to advance the lifestyles of numerous societies, especially in Muslim states. The Right Path to Health is health education through religion and health promotion through an Islamic lifestyle. Avoiding contaminated and prohibited foods such as blood, dead animals, and pork are highly appropriate in the Islamic way of living. Eating fruits and vegetables, fish, cooked food, and brief nutrition are emphasized. Chewing food well, slow and calm eating, washing hands before and after meals and cleaning the mouth with Miswak[12], and brushing teeth are strongly recommended in Muslim cultures, which is responsible for the fast healing of wounds. All these habits are part of Muslim culture (Padela and Zaidi 2018).

## 6. Conclusions

Traditional Islamic teachings and practice place a premium on protecting human dignity from harm and destruction. Policies and guidelines for responses, including that of contemporary social and health care, can be derived from the laws and principles that guide the way of life and treatment of Muslims and are primarily referenced in the Quran and the Prophetic traditions. A speedy recovery of wounds and infections is ensured by the Islamic diet, Islamic practices, and the Islamic way of life. If man makes an honest effort to grasp the meaning of the Quran and Sunnah, he will see that it is a comprehensive manual for all aspects of his existence. More study and human intelligence is desperately

needed to decipher the clues left by the Holy Quran and the Sunnah of the Holy Prophet Hazrat Muhammad.

**Author Contributions:** Conceptualization, T.A. and T.A.; methodology, T.A.; software, H.S.; validation, T.A., H.S. and T.A.; formal analysis, T.A.; investigation, T.A.; resources, T.A.; data curation, H.S.; writing—original draft preparation, X.X.; writing—review and editing, X.X.; visualization, X.X.; supervision, T.A.; project administration, H.S.; funding acquisition, T.A. All authors have read and agreed to the published version of the manuscript.

**Funding:** This research received no external funding.

**Conflicts of Interest:** The authors declare no conflict of interest.

## Notes

[1] The life and traditions of the Holy Prophet Hazarat Muhammad (PBUH during his life time from 29 August, 570 CE to 8 June 632CE

[2] It is an Arabic term which means rubbing dry soil to clean the private parts of body after excretion of waste from the human body.

[3] Halal food is the food which is allowed by the Islamic Law; during the initial years many things were halal, but by he passage of time, some animal's meat was considered as haram due to certain reasons.

[4] Some foods in Islam have been categorized as Haram and consumption of such food is illegal in Islam and it is considered as a great sin to eat haram food such as pork, wine etc.

[5] Modern science could not find any problem in the flesh of the donkey but Islam says its unclean in Bukhari, hadith no. 5528

[6] Not all kinds, but some fish can be the cause of pathogenic bacterial hazards. In Islam, only a few kinds of fish are allowed to be eaten, but a large number of fish and whales are not allowed to be eaten.

[7] Rakah are the movements in the Muslim prayers which contains different postures.

[8] Gusal is a kind of bath which is compulsory for a Muslim on certain occasions. It also means purifying one's body from all kinds of filth and its full method has been explained in Islam.

[9] Muslims observe a period of fasting in the Month of Ramadan which is the ninth month for Muslims. In this month, Muslims observe a fasting period of thirty days. For almost 12 h in the day, they do not eat anything.

[10] In Islam, marriage is encouraged as soon as an individual becomes an adult.

[11] To circumcise the male reproductive part is compulsory in Islam and nobody is allowed to marry without circumcision.

[12] A special stick is used to clean the teeth by Muslims which is acquired from a special tree.

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
