# Peer review of "An Islamic Perspective on Infection Treatment and Wound Healing"

_religions, doi:10.3390/rel14081044_

Round 1

Reviewer 1 Report

Thank you to the authors for attempting to shed light on this important topic.

Critical revisions needed:

1. Muslims have traditionally and continue to use means of healing beyond those mentioned the Quran and Sunnah, as long as they are halal means.  The authors make absolutely no mention of this and appear to claim that for Muslims only treatments directly from the Quran and Sunnah are valid for use.  The connection of Islamic jurisprudence here is critical and missing from the article.

2. The authors must clearly define the subject that they are addressing.  The terms 'wound' and 'healing' are not clearly defined and they are used in a variety of ways throughout the paper.  Healing, as used in the Quran also refers to salvation, but this is not mentioned when using these references.  I would suggest creating a table that summarizes the types of wounds and the proposed methods of healing from Quran and sunnah.  This would help the authors clarify their points.

3. The discussion on soul and body is a very superficial analysis and appears to reflect an acceptance of the dichotomous view of Descartes.  This is distinct from mainstream Islamic tradition in which the soul and body are inextricably connected during the life in the dunya where 'death' is defined by when the soul leaves the body.  The holistic view of the connection between body and soul enables Muslims to understand the spiritual diseases may have physical manifestations and physical diseases may have mental manifestations and so on.  This is critical to appreciate when writing about healing and cures from an Islamic perspective.

4. To restrict the methods by which Muslims have attempted to heal wounds to the time period of the Prophet (SAW) is an incomplete approach and does not reflect the practice of generations of Muslims who did NOT restrict their methods of healing as the paper suggests.  Rather, Muslims have consistently used Islamic jurisprudence to assess new and innovative methods of healing and if they were deemed halal, pursued them to the benefit of all.

5. The purpose of Islamic practices may or may not have anything to do with healing of wounds.  The connections proposed by the authors are not well substantiated (e.g. the comparison between prayer and yoga is highly suspect as the impact of a Muslim's prayer is far beyond wound healing and has to do with receiving forgiveness and creating a bond with Allah, these are not present in yoga).  E.g. the authors state that making ablution five times a day protects one's private parts, but these parts are not part of ablution unless ghusl is made.

6. The connection between diet and wound healing is also not clearly delineated but rather conjecture.  Many Muslims can follow eating 'halal' food but still develop diabetes and a myriad of diseases.  The ethics and sunnah of moderation are clearly beneficial for health as well, but how does this directly relate to wound healing?

7. The authors make several claims that are not substantiated.  E.g. line164, honey is the BEST healer for many diseases.  This is not a direct translation of the verse referred to here as the BEST component is not there.  The authors must be more precise in their translations and the implications to support their arguments.  Similarly line 167, how does the Quran certify that honey is the best healer for almost all diseases?  To what diseases does this statement refer and what is the evidence to support this statement?  The fact that honey is antimicrobial does not imply that it heals almost all diseases.

8. Under method 2, the authors quote a hadith in which the Prophet (SAW) forbids cauterization, then in Method 3, the authors need to explain further what cauterization without fire means.  The current description states 'red-hot' metal -- how is this different from what was forbidden (fire)?

9. In line 94, the authors state that meat consumption has been prohibited most of the time but provide no reference to support this statement.  This appears to be inconsistent with Islamic tradition that clearly provides guidance of what meat should be eaten and how it should be properly slaughtered and cleaned.

10. The scientific connection between what is halal in food and improved wound healing is not based on any comparison.  How can the authors claim that these foods are the best when no comparison is made to any 'non' halal foods.  Further, Muslims are not required to eat all of these foods, but rather they are allowed to choose from among halal options.  This section appears to be highly subjective with little objective evidence to support the claims.

11.  In summary, this paper needs significant revisions before acceptance -- specifically clear definitions of wound and infection and healing.

The quality of the English language is adequate, however, the use of 'proven' and similar words needs to be adjusted to a more humble language such as evidence suggests/supports, etc.

Author Response

Authors’ responses to the Reviewer's comments and requests

We would like to thank the reviewers for their critical and supportive comments regarding our study. All of the suggestions of the Reviewers have been accepted and added to the revised manuscript (see further comments below). Furthermore, in addition to the reviewer’s comments, the manuscript has been revised heavily in terms of improving the language and technical quality of the manuscript and you will see the difference in the revised manuscript indicated with colored text.

Reviewer#1

Comment#1. Muslims have traditionally and continue to use means of healing beyond those mentioned the Quran and Sunnah, as long as they are halal means.  The authors make absolutely no mention of this and appear to claim that for Muslims only treatments directly from the Quran and Sunnah are valid for use.  The connection of Islamic jurisprudence here is critical and missing from the article.

Reply: It is mentioned in the introduction that “To heal the wounds of Muslims, Muslims have traditionally used a variety of methods. Several practices were common in Arabian civilization prior to the origination of Islam, but Islam only endorsed a small number of practices, and the Holy Prophet of Islam focused on specific techniques for treating wounds. Muslims throughout history employed various ways for healing wounds that were not mentioned in the Holy Quran and Sunnah, yet the Quran and Sunnah had established the guidelines for halal and haram.

Comment#2. The authors must clearly define the subject that they are addressing.  The terms 'wound' and 'healing' are not clearly defined and they are used in a variety of ways throughout the paper.  Healing, as used in the Quran also refers to salvation, but this is not mentioned when using these references.  I would suggest creating a table that summarizes the types of wounds and the proposed methods of healing from Quran and sunnah.  This would help the authors clarify their points.

Reply: The terms were clearly defined in the introduction part and A new table was incorporated before the explanation  of methods of treatment.

Comment#3. The discussion on soul and body is a very superficial analysis and appears to reflect an acceptance of the dichotomous view of Descartes.  This is distinct from mainstream Islamic tradition in which the soul and body are inextricably connected during the life in the dunya where 'death' is defined by when the soul leaves the body.  The holistic view of the connection between body and soul enables Muslims to understand the spiritual diseases may have physical manifestations and physical diseases may have mental manifestations and so on.  This is critical to appreciate when writing about healing and cures from an Islamic perspective.

Reply: The human soul is the software, while the body is the hardware. The soul is not a tangible thing, yet it does reside in the body. Islam emphasizes that the soul is the most significant and ever-living thing, whereas the body is only ephemeral. As software is crucial for a computer's performance, so is the soul for an individual's existence. It was further elaborated according to the directions of the reviewer.

Comment #4. To restrict the methods by which Muslims have attempted to heal wounds to the time period of the Prophet (SAW) is an incomplete approach and does not reflect the practice of generations of Muslims who did NOT restrict their methods of healing as the paper suggests.  Rather, Muslims have consistently used Islamic jurisprudence to assess new and innovative methods of healing and if they were deemed halal, pursued them to the benefit of all.

Reply: This study focuses on the fundamental Islamic practices for treating wounds. This time period encompasses the revelation of the holy Quran as well as the sunnah of the holy Prophet. The approaches of Islamic physicians like Abu Ali Sina and Islamic jurisprudence are vast subjects that needs a long discussion and many research articles can be written in that area. This research paper examines the key Islamic methodologies used by Muhammad in accordance with the Holy Quran.

Comment#5. The purpose of Islamic practices may or may not have anything to do with healing of wounds.  The connections proposed by the authors are not well substantiated (e.g. the comparison between prayer and yoga is highly suspect as the impact of a Muslim's prayer is far beyond wound healing and has to do with receiving forgiveness and creating a bond with Allah, these are not present in yoga).  E.g. the authors state that making ablution five times a day protects one's private parts, but these parts are not part of ablution unless ghusl is made.

Reply: I merely likened Yoga to Muslim prayers from the standpoint of physical exercise. In the same way that yoga is a great physical workout, so are Muslim prayers. Yoga is analogous to Muslim prayers in one way, as I have explained in this work.  Ablutions are required before every prayer, and Muslims cleanse their private areas in their ablution. Ghusal requires Muslims to wash their entire body, whereas ablution is the process of cleansing and washing private areas. Fungus is kept at bay by washing private areas.

Comment#6. The connection between diet and wound healing is also not clearly delineated but rather conjecture.  Many Muslims can follow eating 'halal' food but still develop diabetes and a myriad of diseases.  The ethics and sunnah of moderation are clearly beneficial for health as well, but how does this directly relate to wound healing?

Reply: In fact, a powerful immune system is the prerequisite of healing of wounds and powerful immune system cannot be built without balanced diet. In this way we can say that diet indirectly supports healing. Secondly some food items like honey, Ginger, black cumin and many other food items directly improve the healing of wounds. Diabetes is caused when excessive use of diet is made while Islam has stressed on moderation which prevents excessive use.

Comment#7. The authors make several claims that are not substantiated.  E.g. line164, honey is the BEST healer for many diseases.  This is not a direct translation of the verse referred to here as the BEST component is not there.  The authors must be more precise in their translations and the implications to support their arguments.  Similarly line 167, how does the Quran certify that honey is the best healer for almost all diseases?  To what diseases does this statement refer and what is the evidence to support this statement?  The fact that honey is antimicrobial does not imply that it heals almost all diseases.

Reply: According to the Holy Quran, honey may cure all ailments. According to the Quran, honey is the best remedy for many diseases, and it is specifically referenced in the Quran. As a result, we may conclude that honey is an excellent healer. The term best is not specified, but it has been thought to be the cure for all ailments; if an item can give cure for many diseases at the same time, we may deduce that it is a greatest healer. The holy Prophet also emphasized the value of honey for healing, and according to Islamic sayings, it is the best healer.

According to the Quran and the teachings of Prophet Muhammad, peace be upon him, honey has medicinal qualities. The following points support the assertion that "Honey is the best healer according to Islam":

  1. 1. Allah states in Surah An-Nahl (Chapter 16), Verse 69, "Then eat from all the fruits and follow the ways your Lord has laid down [for you]." A liquid of varied colors comes from their guts, bringing humans healing. Indeed, that is an indication of a people that think." This sonnet emphasizes the medicinal qualities of many fruits and their extracts, one of which is honey.
  2. 2. The Prophet Muhammad, peace be upon him, likewise stressed the therapeutic properties of honey. "Use the two remedies: honey and the Quran," he said.
  3.  Because of its low pH and high sugar content, honey is believed to have antibacterial qualities that hinder the development of dangerous germs. It also includes enzymes that create hydrogen peroxide, which has antibacterial properties. These qualities contribute to honey's ability to heal wounds and prevent infection.
  4.  Honey has been used for ages in Islamic traditional medicine due to its therapeutic effects. It has been used to cure a variety of diseases, including digestive disorders, respiratory troubles, wounds, burns, and skin concerns.

Comment #8. Under method 2, the authors quote a hadith in which the Prophet (SAW) forbids cauterization, then in Method 3, the authors need to explain further what cauterization without fire means.  The current description states 'red-hot' metal -- how is this different from what was forbidden (fire)?

Reply: Islam acknowledges and supports the use of medical interventions to heal and treat illnesses. Cauterization, which involves the use of heat to treat wounds, is allowed in Islam as it falls under the broader category of seeking medical treatment. There are several hadiths that mention cauterization as a recommended method of treatment. For example, it is reported that Prophet Muhammad, peace be upon him, said, "There is healing in three: in the cutting of a vein, in cauterizing with fire, and in the application of a cupping glass, but I do not endorse cauterizing." This hadith acknowledges cauterization as a valid method of treatment, but it should be noted that Prophet Muhammad personally did not prefer it as a method.

Comment#9. In line 94, the authors state that meat consumption has been prohibited most of the time but provide no reference to support this statement.  This appears to be inconsistent with Islamic tradition that clearly provides guidance of what meat should be eaten and how it should be properly slaughtered and cleaned.

Reply: Prophet Muhammad, peace be upon him, led by example and often practiced moderation in his own diet. He would sometimes go for extended periods without consuming meat, opting for simpler foods and occasionally eating meat on special occasions. It is important to note that Islam does not prohibit meat consumption entirely but encourages Muslims to be mindful of their dietary choices, promoting moderation, and reminding individuals not to excessively indulge in any particular food item, including meat. This teaching about not consuming meat too frequently aligns with the overall focus on balance, self-control, and maintaining good health in both body and spirit. A reference was also provided to substantiated the argument that excessive meat consumption is prohibited in Islam.

Comment#10. The scientific connection between what is halal in food and improved wound healing is not based on any comparison.  How can the authors claim that these foods are the best when no comparison is made to any 'non' halal foods.  Further, Muslims are not required to eat all of these foods, but rather they are allowed to choose from among halal options.  This section appears to be highly subjective with little objective evidence to support the claims.

Reply: Halal food especially halal meat is considered less toxicant as compared to Pork which is haram in Islam. Pork meat is more toxicant in nature which contains huge fat in it. The main point in the paper has been made is that halal meat is considered as less toxicant which can increase the immunity of individual when it is consumed while haram meat is more toxicant which can be harmful for health.

Comment#11.  In summary, this paper needs significant revisions before acceptance -- specifically clear definitions of wound and infection and healing

Reply: In the introduction part in the line 73 Wounds, infections and healing has been defined and muslim way of treating wounds have also been introduced.

Reviewer 2 Report

This essay could make a potentially valuable contribution to not only the specific topic of Islamic contributions to health and healing; there is no doubt about the value of Islamic (Qur’anic and Holy Books)-based healing, but it could with some substantial revisions, also contribute to wider fields of anthropology of religion and medical anthropology. However, as it now stands in this version of the essay, there are some problems that need more careful reflection and relating to social contexts of use (perhaps a few brief social examples, not solely from textual analysis of theology,  would strengthen the author’s points), and also, there is need for attention to how these data contribute to broader issues in the fields of religion and science and for specification. What sort of infections and wounds, exactly? Also there is need for greater contextualization, not solely from theological sources (though these should be retained, provided the redundancy is eliminated).  The descriptive material is somewhat redundant; repeated points could be streamlined and cut down to allow space for more analysis and discussion, in the conclusion at least, of broader implications for understandings of divergences, but also connections and overlaps, between religion and science more generally. There is a large body of literature on this topic in cross-cultural studies of religion and medicine: These data, with greater social contextualization and some additional sources consulted, could counter the false and ethnocentric notion that resistance to biomedical interventions (or uses of some combined “supplementary/complementary” treatments) is the result of ignorance, “superstition”, or stubbornness among some populations. See for example the classic work (1955) of Benjamin Paul re: the need to “think like” people in a community. See also a more recent 2015 work by Robert Thornton, and many others, for broader implications, even beyond Islamic communities. The author needs to point out how their essay builds on and contributes to works showing  the rationality, logic, and efficacy of religion-based healing.  

Overall, English is clear and no serious problems that could not be fixed with spell-check and/or proofreading.

Author Response

Authors’ responses to the Reviewer's comments and requests

We would like to thank the reviewers for their critical and supportive comments regarding our study. All of the suggestions of the Reviewers have been accepted and added to the revised manuscript (see further comments below). Furthermore, in addition to the reviewer’s comments, the manuscript has been revised heavily in terms of improving the language and technical quality of the manuscript and you will see the difference in the revised manuscript indicated with colored text.

Reviewer#2

Comment #1 This essay could make a potentially valuable contribution to not only the specific topic of Islamic contributions to health and healing; there is no doubt about the value of Islamic (Qur’anic and Holy Books)-based healing, but it could with some substantial revisions, also contribute to wider fields of anthropology of religion and medical anthropology. However, as it now stands in this version of the essay, there are some problems that need more careful reflection and relating to social contexts of use (perhaps a few brief social examples, not solely from textual analysis of theology,  would strengthen the author’s points), and also, there is need for attention to how these data contribute to broader issues in the fields of religion and science and for specification. What sort of infections and wounds, exactly? Also, there is need for greater contextualization, not solely from theological sources (though these should be retained, provided the redundancy is eliminated).

Reply: This paper discusses wounds like Incised Wound, Laceration, Puncture Wound, Abrasion, Avulsion, Gunshot Wound, Burn, Pressure Ulcer, Surgical Wound, Bite Wound. It also discusses the infections like Bacterial and fungal infections which are associated with wounds.

Comment#2 :The descriptive material is somewhat redundant; repeated points could be streamlined and cut down to allow space for more analysis and discussion, in the conclusion at least, of broader implications for understandings of divergences, but also connections and overlaps, between religion and science more generally.

Reply: Maximum effort was made to avoid redundancy in the whole paper according to the directions of the reviewer in the paper.

Comment #3: There is a large body of literature on this topic in cross-cultural studies of religion and medicine: These data, with greater social contextualization and some additional sources consulted, could counter the false and ethnocentric notion that resistance to biomedical interventions (or uses of some combined “supplementary/complementary” treatments) is the result of ignorance, “superstition”, or stubbornness among some populations. See for example the classic work (1955) of Benjamin Paul re: the need to “think like” people in a community. See also a more recent 2015 work by Robert Thornton, and many others, for broader implications, even beyond Islamic communities.

Ans: The Classic works of Benjamin Paul and Robert Thornton to improve the paper further.

Comment#4: The author needs to point out how their essay builds on and contributes to works showing  the rationality, logic, and efficacy of religion-based healing.

Reply: we tried to point out the contribution of this essay on the study of religion based healing because it presents the perspective of worlds second biggest  religion about healing of wounds.

Round 2

Reviewer 1 Report

The authors have addressed reviewer concerns and improved the quality and coherency of their manuscript significantly.  

Minor suggestions:

1. Lines 115-118 - the authors continue to state that ritual ablution includes cleaning of private parts.  This is factually incorrect.  Perhaps there is a different understanding of private parts?  If one defines private parts as the awrah, then the steps of ritual ablution (wudu defined as washing of hands, mouth, nose, face, arms, head, neck, and feet) do NOT require cleaning these parts each time someone prays.  The broader concept of taharah does include cleaning of private parts after urination/defecation and that this level of cleanliness is required for prayer.  This may appear what was intended, but the current text needs to be more accurate.

 2. Lines 267-270 - a standard definition of : To cauterize is to seal off a wound or incision by burning it or freezing it, usually with a hot iron, electricity, or chemicals.  In the Hadith, the Prophet (saw) explicitly forbids using branding with fire as a method of cauterization.  For clarity,  I would suggest that the authors add ", but not with fire" to the end of the first sentence of this paragraph.  I do appreciate that this is stated later in the paragraph.

No additional comments.

Author Response

Authors’ responses to the Reviewer's comments and requests

We would like to thank the reviewers for their critical and supportive comments regarding our study. All of the suggestions of the Reviewers have been accepted and added to the revised manuscript (see further comments below). Furthermore, in addition to the reviewer’s comments, the manuscript has been revised heavily in terms of improving the language and technical quality of the manuscript and you will see the difference in the revised manuscript indicated with colored text.

Reviewer#1

. Lines 115-118 - the authors continue to state that ritual ablution includes cleaning of private parts.  This is factually incorrect.  Perhaps there is a different understanding of private parts?  If one defines private parts as the awrah, then the steps of ritual ablution (wudu defined as washing of hands, mouth, nose, face, arms, head, neck, and feet) do NOT require cleaning these parts each time someone prays.  The broader concept of taharah does include cleaning of private parts after urination/defecation and that this level of cleanliness is required for prayer.  This may appear what was intended, but the current text needs to be more accurate.

Answer: The suggestion was followed in line no 118 and 119 and the text was replaced.

  1. Lines 267-270 - a standard definition of : To cauterize is to seal off a wound or incision by burning it or freezing it, usually with a hot iron, electricity, or chemicals. In the Hadith, the Prophet (saw) explicitly forbids using branding with fire as a method of cauterization. For clarity,  I would suggest that the authors add ", but not with fire" to the end of the first sentence of this paragraph.  I do appreciate that this is stated later in the paragraph.

Answer: The suggestion of the reviewer was followed.

Reviewer 2 Report

For the most part, I the essay was improved, very interesting, and issues raised earlier such as redundancy were addressed and mostly (though not entirely) resolved. There remains the need (as  I also pointed out in the  review form ) to elaborate a bit more on the broader implications in terms of what "science" (western and/or Islamic) means in light of the data presented here. Also, the essay still needs to be proofread for a few remaining typographic errors. For the revised version, I recommend "Accept with minor revisions"---an improved, very interesting essay but still could be enriched analytically and polished stylistically.

The essay still needs to be proofread for a few remaining typographic errors.

Author Response

Authors’ responses to the Reviewer's comments and requests

We would like to thank the reviewers for their critical and supportive comments regarding our study. All of the suggestions of the Reviewers have been accepted and added to the revised manuscript (see further comments below). Furthermore, in addition to the reviewer’s comments, the manuscript has been revised heavily in terms of improving the language and technical quality of the manuscript and you will see the difference in the revised manuscript indicated with colored text.

Reviewer#2

For the most part, I the essay was improved, very interesting, and issues raised earlier such as redundancy were addressed and mostly (though not entirely) resolved. There remains the need (as  I also pointed out in the  review form ) to elaborate a bit more on the broader implications in terms of what "science" (western and/or Islamic) means in light of the data presented here. Also, the essay still needs to be proofread for a few remaining typographic errors. For the revised version, I recommend "Accept with minor revisions"---an improved, very interesting essay but still could be enriched analytically and polished stylistically.

Answer: According to the suggestions of the proof-readings were made frequently and followed the suggestions in true spirit.